# Localization of *oskar* mRNA by agglomeration in ribonucleoprotein granules

**Catherine E. Eichler** ©, **Hui Li** ©, **Michelle E. Grunberg** ©, **Elizabeth R. Gavis** ©*

Department of Molecular Biology, Princeton University, Princeton, New Jersey, United States of America

☯ These authors contributed equally to this work.
* gavis@princeton.edu

**Data Availability Statement:** All relevant data are within the manuscript and its Supporting Information files.

**Funding:** This work was supported by National Institute of Health (NIH) grant R35 GM126967 to

## Abstract

Localization of *oskar* mRNA to the posterior of the *Drosophila* oocyte is essential for abdominal patterning and germline development. *oskar* localization is a multi-step process involving temporally and mechanistically distinct transport modes. Numerous cis-acting elements and trans-acting factors have been identified that mediate earlier motor-dependent transport steps leading to an initial accumulation of *oskar* at the posterior. Little is known, however, about the requirements for the later localization phase, which depends on cytoplasmic flows and results in the accumulation of large *oskar* ribonucleoprotein granules, called founder granules, by the end of oogenesis. Using super-resolution microscopy, we show that founder granules are agglomerates of smaller *oskar* transport particles. In contrast to the earlier kinesin-dependent *oskar* transport, late-phase localization depends on the sequence as well as on the structure of the spliced *oskar* localization element (SOLE), but not on the adjacent exon junction complex deposition. Late-phase localization also requires the *oskar* 3′ untranslated region (3′ UTR), which targets *oskar* to founder granules. Together, our results show that 3′ UTR-mediated targeting together with SOLE-dependent agglomeration leads to accumulation of *oskar* in large founder granules at the posterior of the oocyte during late stages of oogenesis. In light of previous work showing that *oskar* transport particles are solid-like condensates, our findings indicate that founder granules form by a process distinct from that of well-characterized ribonucleoprotein granules like germ granules, P bodies, and stress granules. Additionally, they illustrate how an individual mRNA can be adapted to exploit different localization mechanisms depending on the cellular context.

## Author summary

Many events that occur early in animal development, including formation of the reproductive cells, require production of proteins at the particular cellular locations where they function. Localization of the messenger RNAs (mRNAs) that code for these proteins is an effective and widespread method to accomplish such on-site translation. Localization of *oskar* mRNA to the posterior end of the *Drosophila* oocyte is essential to produce Oskar protein that in turn directs formation of both the reproductive cells and the abdomen once the embryo begins to develop. Previous studies have shown that *oskar* mRNA is

ERG. CRE was supported by NIH training grant T32 GM007388. HL was supported by a pre-doctoral fellowship jointly funded by the China Scholarship Council and Princeton University. The funders had no role in study design, data collection and analysis, decision to publish, or preparation of the manuscript.

**Competing interests:** The authors have declared that no competing interests exist.

assembled along with proteins into particles that are transported to the posterior of the oocyte by molecular motors. After this initial localization, *oskar* continues to accumulate at the posterior when motor-dependent transport is no longer possible. Here we show that this later phase of *oskar* localization occurs by an unexpected mechanism in which many smaller *oskar* particles "stick" together, forming large granules that can contain tens to hundreds of *oskar* molecules. We identify features of *oskar* mRNA that control this granule formation and show how they differ from those that control the earlier motor-dependent transport. Our results reveal greater diversity of mRNA localization mechanisms and show how an individual mRNA can be adapted to use different mechanisms as needed.

## Introduction

Localization of mRNAs to subcellular domains plays an important role in generating morphological and functional asymmetry through local protein production. Localization of *oskar* (*osk*) mRNA to the posterior of the *Drosophila* oocyte and its on-site translation are essential for formation of the germ plasm, the specialized embryonic cytoplasm required for abdominal patterning and germ cell formation during embryogenesis [1–4]. The trafficking of mRNAs to different subcellular locations is controlled by cis-acting RNA sequences and/or structures that most commonly reside within 3′ untranslated regions (3′ UTRs), and proteins that recognize these localization elements. These proteins may interface with cellular transport machinery or they may facilitate association with cellular structures or organelles, including phase transitioned condensates [5, 6]. Multiple cis-acting elements along with a coterie of trans-acting factors contribute to the multi-step process of *osk* localization throughout oogenesis [7, 8].

*Drosophila* oogenesis proceeds through 14 morphologically defined stages; during the first 10 stages the oocyte develops within an egg chamber, accompanied by 15 sister nurse cells that support oocyte growth and development by providing maternal mRNAs such as *osk* as well as proteins and organelles. *osk* is transported from the nurse cells to the oocyte by dynein; this transport requires sequence elements and structures in the 3′ UTR [9, 10]. One of these, called the oocyte entry signal (OES), links *osk* mRNA to dynein by binding to the RNA-binding protein Egalitarian and the adaptor Bicaudal-D [10–12]. Throughout stages 8 to 10, *osk* travels to the posterior of the oocyte by kinesin-mediated transport [13, 14]. This transport requires an unusual localization element, a stem-loop structure called the Spliced *oskar* Localization Element (SOLE) [15]. The SOLE comprises the last 18 nucleotides of the first *osk* exon and the first 10 nucleotides of the second *osk* exon and therefore splicing of the first *osk* intron is crucial for localization. Mutational analysis showed that SOLE function depends on its structural integrity rather than on its sequence. Splicing of the first *osk* intron also plays a role in *osk* localization through recruitment of the exon junction complex (EJC) [15]. The EJC/SOLE functions as a unit to activate kinesin motility, leading to accumulation of *osk* at the posterior [15, 16].

In addition to its role in nurse cell-to-oocyte transport, the *osk* 3′ UTR contributes in multiple ways to posterior transport within the oocyte [17]. In the oocyte, *osk* multimerizes, forming ribonucleoprotein particles (RNPs) containing up to 4 *osk* mRNAs [18]. The RNA-binding protein Bruno 1 (Bru1) binds to the *osk* 3′ UTR and promotes *osk* oligomerization to form translationally repressed complexes [19, 20]. Bru1 drives assembly of *osk* transport RNPs through phase separation; these *osk* RNP condensates mature to a solid state, which is required for their transport, posterior accumulation, and translation [19]. In addition, a short sequence

at the tip of the OES promotes *osk* dimerization, allowing transgenic reporter mRNAs with this dimerization domain, but lacking the SOLE, to "hitchhike" to the posterior on endogenous wild-type *osk* mRNA [21, 22]. Whether hitchhiking contributes to localization of endogenous *osk* remains to be determined, however.

The *osk* 3′ UTR also contains binding sites for the double-stranded RNA-binding protein Staufen (Stau) [23, 24]. Stau associates with *osk* RNPs upon their entry to the oocyte and remains colocalized with *osk* at the posterior of the oocyte [1, 2, 18, 25, 26]. Stau facilitates kinesin-dependent transport [13, 23, 25] and is also required for *osk* mRNA translation and anchoring at the posterior [26]. Additionally, Osk protein itself is required to maintain *osk* mRNA at the posterior of the oocyte [4, 27].

At the posterior, Osk initiates assembly of the germ plasm and nucleates formation of RNP condensates called germ granules [8]. Numerous mRNAs become enriched in the germ plasm through their incorporation into germ granules, which ensures their inheritance by the germ cell progenitors, called pole cells, during embryogenesis. Among these is *nanos* (*nos*), whose translation depends on Osk and is crucial for both abdominal patterning and pole cell development [28–31]. For many germ granule mRNAs including *nos*, elements contained within 3′ UTRs are sufficient to direct their germ granule localization [32–34].

At the end of stage 10 of oogenesis, the nurse cells extrude their contents en masse into the oocyte and degenerate. Just prior to this nurse cell dumping, a bulk cytoplasmic flow called ooplasmic streaming is initiated that mixes the oocyte and incoming nurse cell contents [35, 36]. Notably, additional *osk* mRNA enters the oocyte and accumulates at the posterior during this later period, propelled by nurse cell dumping and ooplasmic streaming rather than by motor-dependent transport [37, 38]. This late phase of *osk* accumulation amplifies the germ plasm, allowing production of additional Osk protein and enlargement of germ granules [38–40]. Since the quantity of germ plasm formed, and consequently the quantities of the abdominal determinant Nos and germ cells produced in the embryo, depend directly on the amount of *osk* mRNA localized during oogenesis [30, 41, 42], this amplification is important for abdominal patterning and reproduction. Failure to accumulate *osk* mRNA and protein at late stages of oogenesis results in loss of abdominal segments and pole cells [38, 39].

By the end of oogenesis, posteriorly localized *osk* resides in large granules containing up to 250 *osk* mRNAs [18]. These granules ultimately mediate the degradation of *osk* mRNA in the embryo, where it is toxic to pole cells, and we have referred to them as founder granules to distinguish them from germ granules [43]. How *osk* transport RNPs are organized into larger founder granules has not been investigated. Moreover, which features of *osk* mediate the late phase of localization and whether they are distinct from the signals that mediate the earlier microtubule-dependent steps remain unknown. Using a combination of quantitative confocal imaging and qualitative stimulated emission depletion (STED) microscopy, we show that founder granules are agglomerations of smaller *osk* transport RNPs. By analyzing the behavior of *osk* transgenes with mutations or deletions that affect the *osk* SOLE and 3′ UTR, we show that, in contrast to localization during stages 8 to 10, both the structure and the sequence of the SOLE, but not the nearby exon junction complex (EJC) deposition, are necessary for late-phase localization. Late-phase localization also requires the *osk* 3′ UTR and by swapping the *nos* and *osk* 3′ UTRs, we show that each is sufficient to target mRNA to the appropriate granule. Together, our results demonstrate that 3′ UTR-mediated targeting together with SOLE-dependent agglomeration result in accumulation of *osk* in large founder granules at the posterior of the oocyte during late stages of oogenesis.

## Results

### Founder granules in late-stage oocytes are agglomerates of *osk* RNPs

To examine how *osk* accumulates in founder granules, we performed a spatiotemporal analysis of wild-type *osk* mRNA, visualizing *osk* in both nurse cells and anterior region of the oocyte from stage 8 to stage 9 (S1 Fig) and at the posterior of the oocyte from stage 8 to stage 13 by single molecule fluorescence in situ hybridization (smFISH) and STED microscopy (Figs S1 and 1). Because we were limited to 2D STED, the analysis is qualitative rather than quantitative. From stage 8 to stage 10, diffraction-limited confocal spots resolve into individual, largely round puncta consistent with *osk* transport RNPs. These particles accumulate at high density at the anterior oocyte cortex at stage 8 and the posterior cortex at stages 8 to 10 but remain distinct from each other in both confocal and STED images (Figs S1 and 1). By stage 12, as the late phase of localization progresses, individual confocal spots often resolve to multiple puncta and by stage 13, they appear to comprise agglomerates of many individual *osk* RNPs, independent of probe concentration (Figs 1 and S2A). This is congruent with the dramatic increase in mRNA content of localized *osk* granules that occurs during the later stages of oogenesis [18]. By contrast, lipid droplets, which are similar in size to founder granules, appear homogeneous (S2B Fig).

The ability to resolve individual *osk* RNPs within founder granules indicates a different organization from germ granules. In contrast to *osk*, germ granule mRNAs travel within the

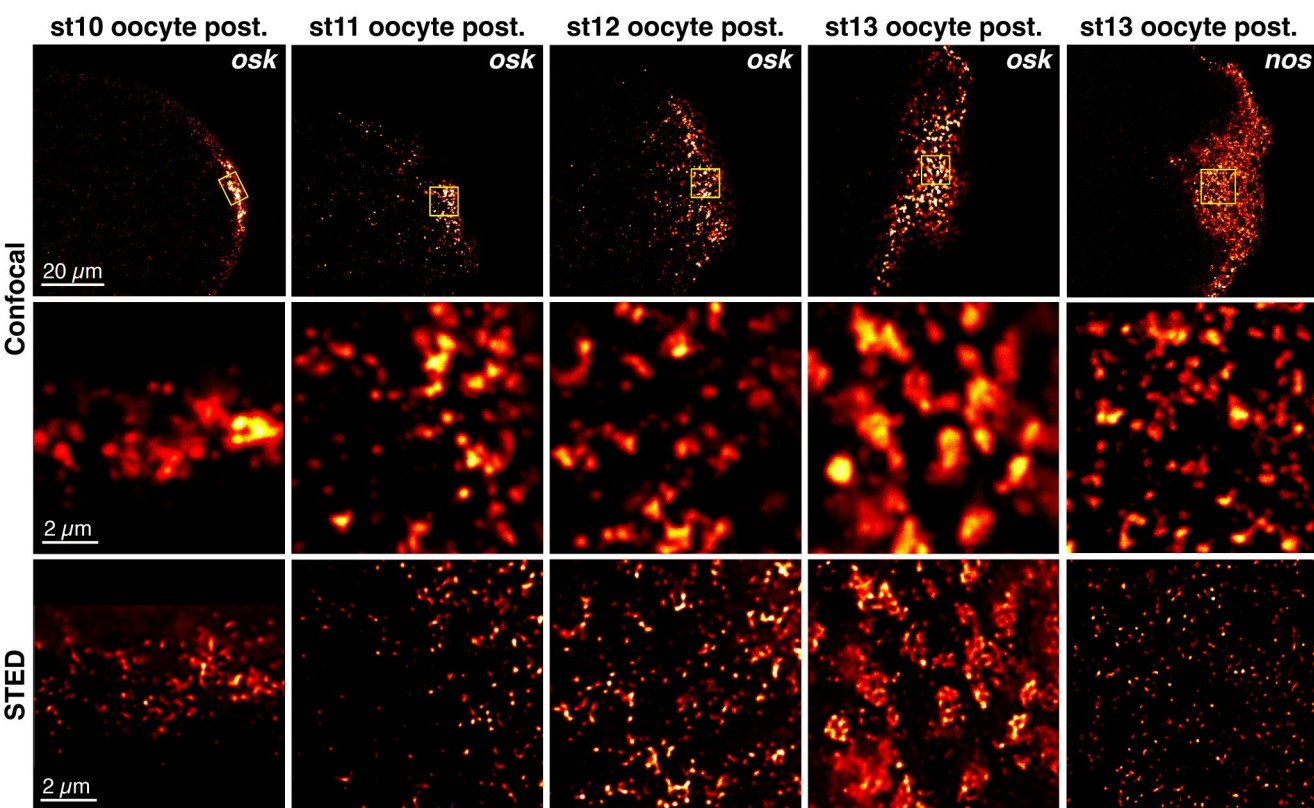

**Fig 1. Founder granules comprise agglomerates of *osk* RNPs.** The top row shows single confocal sections of the posterior region of oocytes at stages 10 to 13, with *osk* or *nos* detected by smFISH. The yellow boxes indicate ROIs imaged using STED microscopy as shown in the panels in the bottom row. Panels in the middle row show confocal images of each ROI prior to STED for comparison. Images are rendered using the Red Hot lookup table in Fiji and scale bars are indicated.

oocyte in RNPs containing only a single transcript. Within the germ plasm, these RNPs become incorporated into phase-separated protein scaffolds nucleated by Osk. Upon incorporation, mRNAs self-associate to form homotypic clusters containing many like transcripts. Distinct clusters of different mRNAs (e.g., *nos* versus *Cyclin B*) can be resolved by super-resolution microscopy and although an individual cluster can contain tens of transcripts, each cluster appears as a single puncta [40, 44]. For direct comparison to *osk* in founder granules, we visualized *nos* using STED microscopy. By confocal microscopy, germ granule associated *nos* appears as bright spots, each corresponding to a germ granule as previously shown [18] and–in contrast to founder granules–these resolve to single puncta with STED (Fig 1). These data suggest that founder granules form by a different mechanism than germ granules, through the agglomeration of RNPs that remain physically distinguishable.

## Late phase *osk* mRNA localization does not appear to require the EJC

We next sought to determine whether elements involved in the active transport of *osk* to the posterior during stages 8 to 10 also function in *osk* localization during late oogenesis. We generated flies expressing a genomic *osk* transgene tagged with a superfolder *gfp* (*sfgfp*) sequence (*osk-sfgfp*), as well as a version with a deletion of the first two introns (*oskΔi1,2-sfgfp*) (Fig 2A). Deletion of intron 1 does not affect SOLE formation but prevents EJC deposition near the exon-exon junction [15]. The *sfgfp* sequence tag allows for detection of the transgenic mRNA by smFISH in the context of endogenous *osk* mRNA. Since the *oskΔi1,2-sfgfp* transgenic mRNA is not expected to localize on its own by kinesin-dependent transport, we reasoned that the presence of endogenous *osk* mRNA and consequent production of Osk protein would be necessary for the initial establishment of germ plasm and for the retention of *osk* mRNA during late stages of oogenesis. The *osk-sfgfp* and *oskΔi1,2-sfgfp* were expressed at comparable levels as determined by RT-qPCR (S3 Fig).

Quantification of the total localized *sfgfp* fluorescence intensity in confocal sections through the entire germ plasm at stage 10 showed that in the presence of wild-type *osk*, both the *osk-sfgfp* and *oskΔi1,2-sfgfp* mRNAs can localize during mid-oogenesis (Fig 2B–2E and 2H). However, localization of the *oskΔi1,2-sfgfp* mRNA is less efficient than the *osk-sfgfp* mRNA (Fig 2D, 2E and 2H). This is consistent with the finding that without EJC deposition on *osk* mRNA, transport efficiency of *osk* RNPs is reduced [15] and suggests that the *oskΔi1,2-sfgfp* mRNA only localizes by hitchhiking with endogenous *osk* using the dimerization domain in the 3′ UTR [22]. By stage 13, the localized fluorescence intensities of the *osk-sfgfp* and *oskΔi1,2-sfgfp* mRNAs increase by 34% and 43%, respectively (Fig 2F–2H). Therefore, deposition of the EJC near the first exon-exon junction is not required for late-phase *osk* localization.

## Late-phase *osk* localization requires the SOLE UA-rich proximal stem sequence

Although the *oskΔi1,2-sfgfp* mRNA is not bound by the EJC near the first exon-exon junction, it can form the SOLE. To test whether the SOLE influences late-phase *osk* localization independently of the EJC, we generated a version of the *osk-sfgfp* transgene with the SOLE proximal stem sequence substituted by *lacZ* sequence (SOLE^PS-Lz^; Fig 3A). This mutation was previously shown to disrupt SOLE localization activity during mid-oogenesis without affecting EJC deposition [15]. As expected, the *oskSOLE^PS-Lz^-sfgfp* mRNA localizes to the posterior of the oocyte by stage 10 in the presence of endogenous *osk*, although with reduced efficiency compared to the *osk-sfgfp* control mRNA (Fig 3B, 3D and 3J). This is consistent with the reduced localization efficiency at stage 10 of the *oskΔi1,2-sfgfp* mRNA (Fig 2D, 2E and 2H). However, in contrast to the *oskΔi1,2-sfgfp* mRNA, the *oskSOLE^PS-Lz^-sfgfp* mRNA is not further enriched by

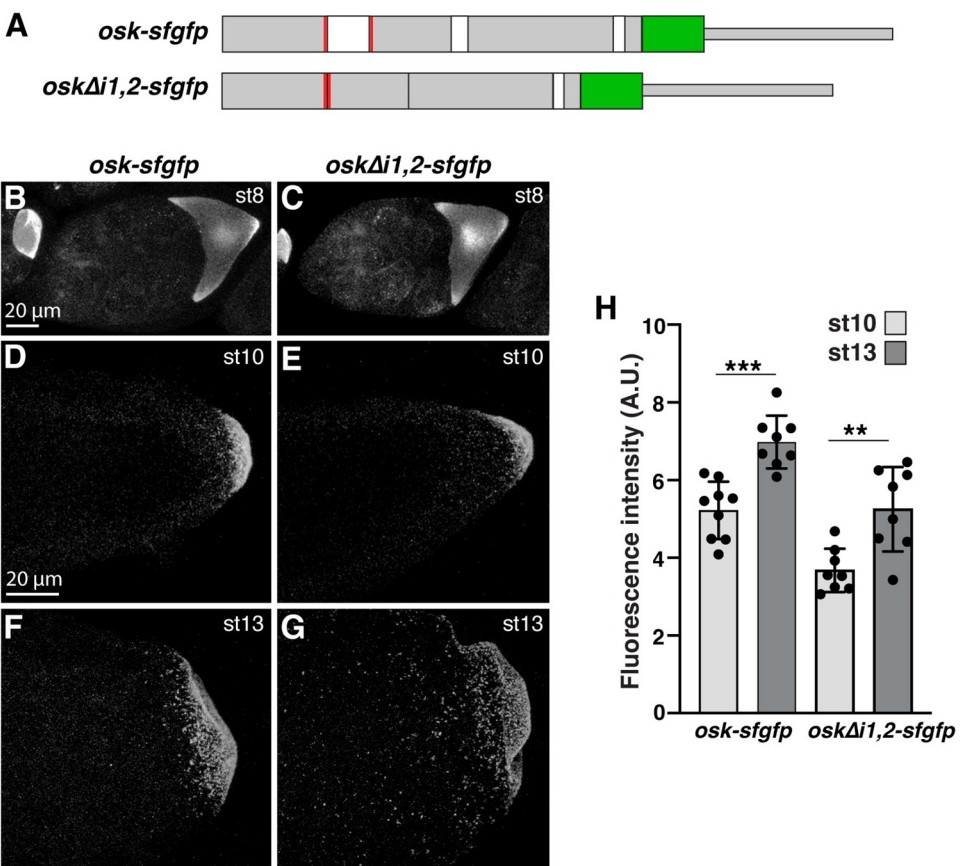

**Fig 2. The EJC is not required for late-phase *osk* localization.** (A) Structure of *osk-sfgfp* and *oskΔi1,2-sfgfp* transgenes. Grey boxes: *osk* exonic sequences; white boxes: *osk* introns; red bars: sequences creating the SOLE; green boxes: *sfgfp*; thinner boxes indicate 3′ UTRs. (B-G) Confocal z-series projections of transgenic stage 8 egg chambers (B, C), stage 10 oocytes (D, E), and stage 13 oocytes (F, G). Anterior is toward the left. The entirety of the germ plasm was captured. Transgenic mRNAs were detected by smFISH using probes for *sfgfp* labeled with 647 fluorophore to avoid detecting fluorescence from Osk-sfGFP protein. (H) Quantification of total localized fluorescence signal intensity. n = 8–9 oocytes each. Individual data points and mean ± standard deviation are shown; ** p < 0.01, *** p < 0.001 as determined by Students t-test. Scale bars are indicated. Source data for the graphs in Fig 2H are provided in S1 Data.

stage 13 (Fig 3C, 3E and 3J). Therefore, the SOLE is required for late-phase *osk* localization, and this role is independent of adjacent EJC deposition.

The earlier phase of *osk* localization relies on the structure, but not the sequence of the SOLE proximal stem [15]. To determine whether this is also the case for the late phase of *osk* localization, we generated a new mutation that disrupts base-pairing of the proximal stem and is thus predicted to disrupt the regular helical structure of the SOLE ($SOLE^{UA-mut}$) [45] as well as a compensatory mutation ($SOLE^{UA-mut\text{-}comp}$) that would restore the proximal stem base-pairing, but not the sequence (Fig 3A). Like the *oskSOLE^{PS-Lz}-sfgfp* mRNA, *oskSOLE^{UA-mut}-sfgfp* mRNA shows reduced localization efficiency at stage 10 in the context of endogenous *osk* and fails to enrich further at stage 13 (Fig 3F, 3G and 3J). The compensatory $SOLE^{UA-mut\text{-}comp}$ mutation restores localization efficiency at stage 10, but does not restore late phase localization (Fig 3H–3J). This is in contrast to the earlier phase of *osk* localization [15]. In all cases, RT-qPCR analysis confirmed that loss of enrichment is not due to low expression of the transgenic mRNA (S3 Fig). Quantification of the number of founder granules containing transgenic mRNA (detected particles containing ≥4 mRNAs, see Materials and Methods) and their size

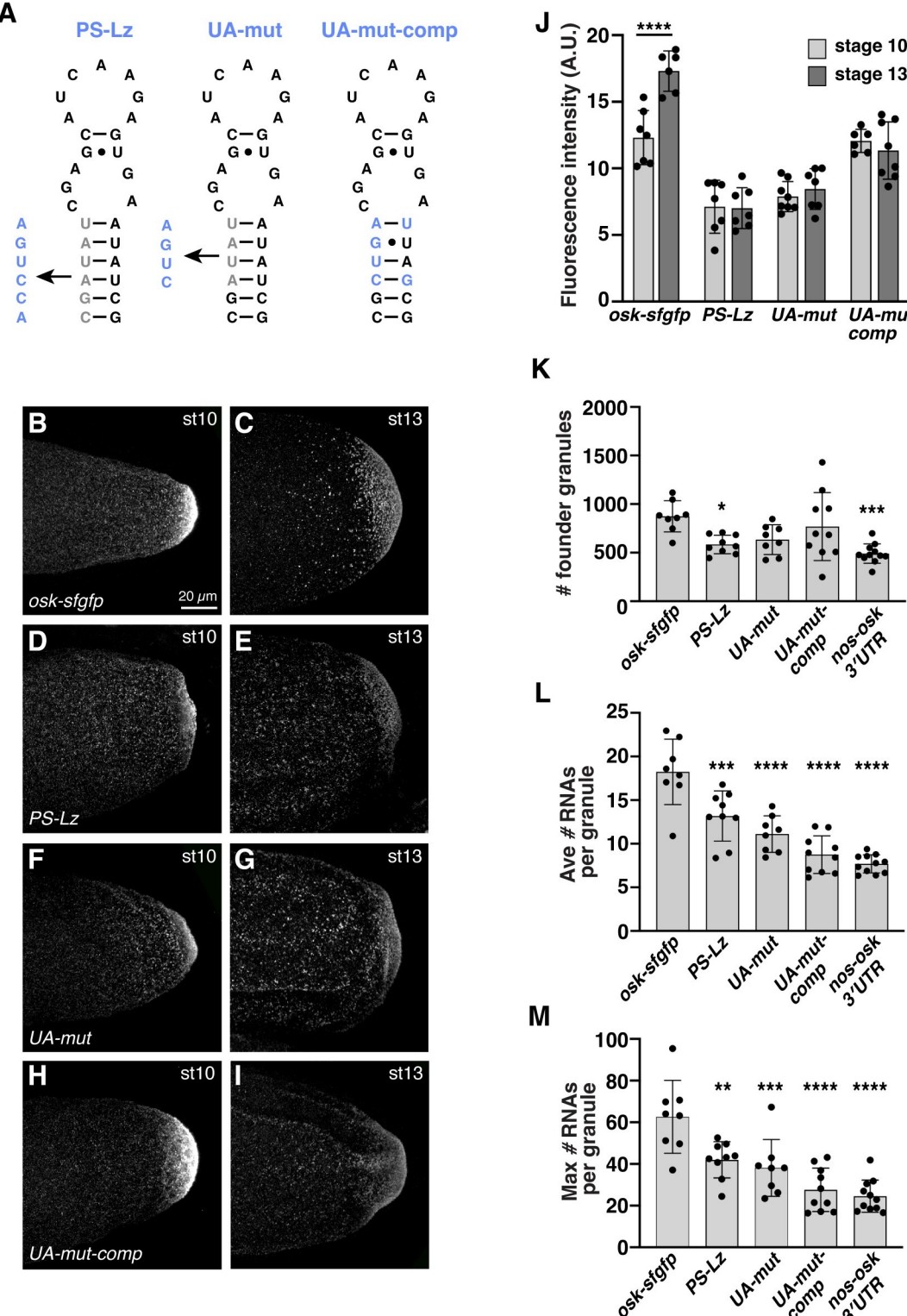

**Fig 3. The sequence and structure of the SOLE is required for late-phase *osk* localization.** (A) The secondary structure of the SOLE with nucleotides of the proximal stem that were changed indicated in gray and the new sequences shown in blue. (B-I) Confocal z-series projections of transgenic stage 10 (B, D, F, H) and stage 13 (C, E, G, I) oocytes, anterior toward the left. The entirety of the germ plasm was captured. Transgenic mRNAs were detected using smFISH probes for *sfgfp*. (J) Quantification of fluorescence intensity of *osk-sfgfp* or *osk-sfgfp* with the indicated SOLE mutations, in the germ plasm from

stages 10 and 13; n = 6–8 oocytes each. (K-M) Quantification of the number of founder granules (particles ≥4 mRNAs) containing the indicated mRNAs (K), or the average (L) and maximum (M) number of mRNAs in those granules; n = 8–11 oocytes each. Similar results were obtained using a requirement of >2 (the minimum for *osk* transport particles [18]). Individual data points and mean ± standard deviation are shown; ** p<0.01, *** p<0.001, **** p<0.0001 as determined by Student's t-test (J) or by one-way ANOVA and Dunnett's post-hoc test (K-M). Scale bars are indicated. Source data for the graphs in Fig 3J-3M are provided in S1 Data.

(i.e., the number of mRNAs per granule) in stage 13 oocytes showed that although none of the mutations significantly affect the number of *osk-sfgfp* containing granules, both the average and maximum number of *osk-sfgfp* in each granule are reduced (Fig 3K–3M). Thus, these data show that both the sequence and the structure of the SOLE are required for late-phase *osk* accumulation.

## SOLE function is required for accumulation of *osk* in founder granules

Our results suggest that the SOLE mutations impair the ability of *osk* to accumulate in founder granules after stage 10. To investigate this possibility, we performed STED microscopy. Similarly to endogenous *osk* at stage 13, *osk-sfgfp* mRNA resides in large granules containing multiple *osk-sfgfp* puncta (Fig 4). By contrast, *oskSOLE^PS-Lz^-sfgfp* mRNA, *oskSOLE^UA-mut^-sfgfp*, and *oskSOLE^UA-mut-comp^-sfgfp* appear largely as individual or small groups of puncta, similar to endogenous *osk* prior to stage 12 (Figs 1, 4 and S1) and consistent with the quantitative analysis of granule size (Fig 3K and 3L). Because wild-type endogenous *osk* is also present and forms founder granules in these oocytes, this pattern suggests that although the SOLE mutant transcripts can localize along with endogenous *osk* prior to nurse cell dumping, they are unable to accumulate in founder granules during late stages of oogenesis. Together, both the quantitative and qualitative data lead us to conclude that SOLE function during late oogenesis is necessary for enrichment of *osk* in the germ plasm by promoting its accumulation in founder granules.

## Late phase *osk* localization requires the *osk* 3′ UTR

In addition to the SOLE, the *osk* 3′ UTR is required for *osk* localization during stages 8 to 10 [17]. Furthermore, the *osk* 3′ UTR of an mRNA lacking the SOLE can facilitate localization by dimerizing with the 3′ UTR of wild-type endogenous *osk*, allowing the SOLE-less mRNA to hitchhike to the posterior [21, 22]. The finding that none of the SOLE mutants tested above support late-phase accumulation indicates that hitchhiking is not occurring at these later stages, however. To test whether the 3′ UTR plays any role in late-phase localization, we generated an *osk-sfgfp* transgene (including the SOLE) in which the *osk* 3′ UTR was replaced by sequences from the *fs(1)K10* 3′ UTR that promote transport of mRNA from the nurse cells to the oocyte, but not localization within the oocyte [46] (Fig 5A). *osk-sfgfp-K10_3′UTR* mRNA is readily detectable by smFISH when expressed under GAL4/UAS control. As expected, *osk-sfgfp-K10_3′UTR* mRNA does not accumulate at the posterior of the oocyte prior to stage 10 even in the presence of wild-type *osk* (Fig 5B). Furthermore, the *osk-sfgfp-K10_3′UTR* does not localize by stage 13 either (Fig 5C), indicating that the *osk* 3′ UTR is required in addition to the SOLE for both earlier and later phases of *osk* localization.

## The *osk* 3′ UTR is sufficient for RNP accumulation in founder granules

Founder granules occupy the germ plasm together with germ granules, but remain physically and functionally distinct. Whereas germ granules are actively incorporated into the pole cells during embryogenesis, founder granules are not and instead recruit degradation machinery to

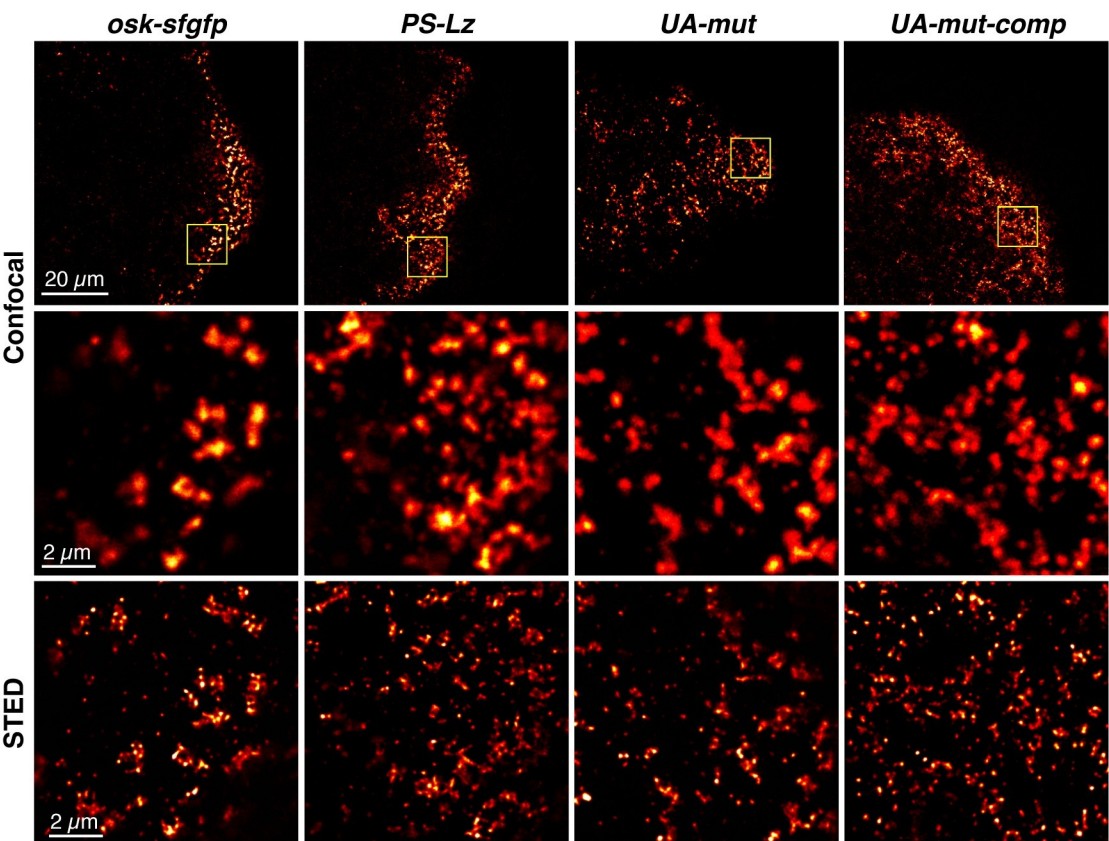

**Fig 4. SOLE mutants affect accumulation of *osk* RNPs in founder granules.** The top row shows single confocal sections of the posterior region of stage 13 oocytes expressing *osk-sfgfp* or *osk-sfgfp* with the indicated SOLE mutations. The transgenic mRNAs were detected by smFISH using probes for *sfgfp*. The yellow boxes indicate ROIs imaged using STED microscopy as shown in the panels in the bottom row. Panels in the middle row show confocal images of each ROI prior to STED for comparison. Images are rendered using the Red Hot lookup table in Fiji and scale bars are indicated.

eliminate *osk* mRNA [18, 43]. As *osk* RNA is toxic to pole cells, its segregation from germ granules is crucial for fertility [43]. Studies of several germ granule mRNAs, including *nos*, *polar granule component*, and *germ cell-less*, have shown that their 3′ UTRs are sufficient for their accumulation in germ granules. To determine if the *osk* 3′ UTR is sufficient to direct mRNA to founder granules, we generated tagged, genomic *osk* and *nos* transgenes with their 3′ UTRs exchanged (Fig 6A) and asked whether the hybrid mRNAs were associated with founder granules or germ granules. The *sfgfp* and *egfp* sequences allowed the respective transgenic mRNAs to be distinguished from endogenous *osk* and *nos* by smFISH. Stau was used as a founder granule marker [43] and *Cyclin B* (*CycB*), an abundant germ granule transcript [18], was used as a germ granule marker. We performed dual smFISH and immunofluorescence to detect the transgenic mRNAs with *sfgfp* or *egfp* probes coupled to one of two far-red fluorophores; germ granules with *CycB* probes coupled to a 565 fluorophore; and founder granules with anti-Stau and a secondary antibody coupled to a 488 fluorophore. The experiments were performed using early embryos, due to the impenetrability of late-stage oocytes to antibodies. We monitored germ granule-association as far-red fluorescence signal (transgenic mRNA) colocalized with 565 fluorescence signal (*CycB*). Because the transgenes encode GFP-labeled proteins, to eliminate any contribution from residual GFP fluorescence we identified founder granules by the criteria they contain mRNA (far-red) and Stau (488) but not *CycB* (565). Based on the

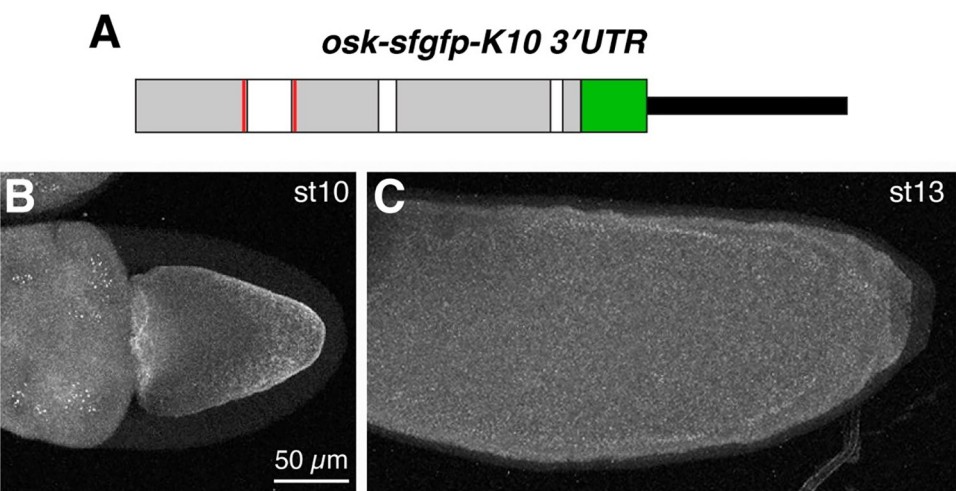

**Fig 5. The *osk* 3′ UTR is required for late-phase localization.** (A) Structure of the *osk-sfgfp-K10_3′UTR* transgene. The *K10* 3'UTR is indicated by the thinner black bar. (B) Confocal z projections of transgenic stage 10 (B) and stage 13 (C) oocytes, with *osk-sfgfp-K10_3′UTR* mRNA detected by smFISH. Anterior is toward the left. Some *osk-sfgfp-K10_3′UTR* mRNA accumulates at the anterior of the oocyte up to stage 10, likely due to the residual localization elements in the *K10* 3'UTR, but the majority is distributed throughout the ooplasm. ≥10 oocytes were imaged for each stage, with similar results. Scale bars are indicated.

previous finding that fewer than 25% of *osk* particles in the germ plasm colocalize with germ granules [18], we set a threshold for granule association at 25% of detected *sfgfp* or *egfp* particles colocalized with the germ granule or founder granule marker as described above. The control *osk-sfgfp* and *egfp-nos* mRNAs behave as expected. 43% of *osk-sfgfp* particles colocalize with Stau (Fig 6B and 6J), whereas colocalization with *CycB* is 22%, below the threshold (Fig 6C and 6J). Conversely, 50% of *egfp-nos* particles colocalize with *CycB* (Fig 6E and 6J), similarly to the behavior of *nos* [18]. In contrast, colocalization of *egfp-nos* with Stau is 6% (Fig 6D and 6J). Similarly to *egfp-nos*, 49% of *osk-sfgfp-nos3′UTR* particles colocalize with *CycB* (Fig 6G and 6J), whereas 23% colocalize with Stau (Fig 6F and 6J). Furthermore, similarly to *osk-sfgfp*, 56% of *egfp-nos-osk3′UTR* particles colocalize with Stau (Fig 6H and 6J), whereas only 10% colocalize with *CycB* (Fig 6I and 6J). Thus, the granule preference of the transgenic mRNAs is determined by the 3′ UTRs. Together with the analysis of *osk-sfgfp-K10_3′UTR* mRNA, these results indicate that the SOLE is not sufficient to target mRNAs to founder granules, rather the *osk* 3′ UTR imparts this function just as the *nos* 3′ UTR directs mRNAs to germ granules.

### The *osk* coding sequences and/or 5′ UTR, but not the SOLE, are required to maintain *osk* mRNA accumulation in late-stage oocytes

Although the *osk* 3′ UTR is sufficient to target *nos* to founder granules, *egfp-nos-osk3′UTR* mRNA is only weakly enriched in the embryonic germ plasm (Fig 7B). To determine whether this results from a defect in the initial localization, presumably by hitchhiking, or a defect in maintenance of *egfp-nos-osk3′UTR* in founder granules over time, we monitored localization over the course of late oogenesis, from stages 10 to 13. In contrast to *osk-sfgfp* mRNA, *egfp-nos-osk3′UTR* does not continue to accumulate at the posterior after stage 10 (Fig 7C–7H and 7L). This behavior is consistent with early phase localization of *egfp-nos-osk3′UTR* by 3′ UTR-mediated hitchhiking but failure to accumulate further due to lack of the SOLE. Moreover, the amount of *egfp-nos-osk3′UTR* mRNA localized at the posterior of the oocyte decreases from stage 12 to stage 13, whereas *osk-sfgfp* mRNA is largely unchanged (Fig 7C–7H and 7L). Both

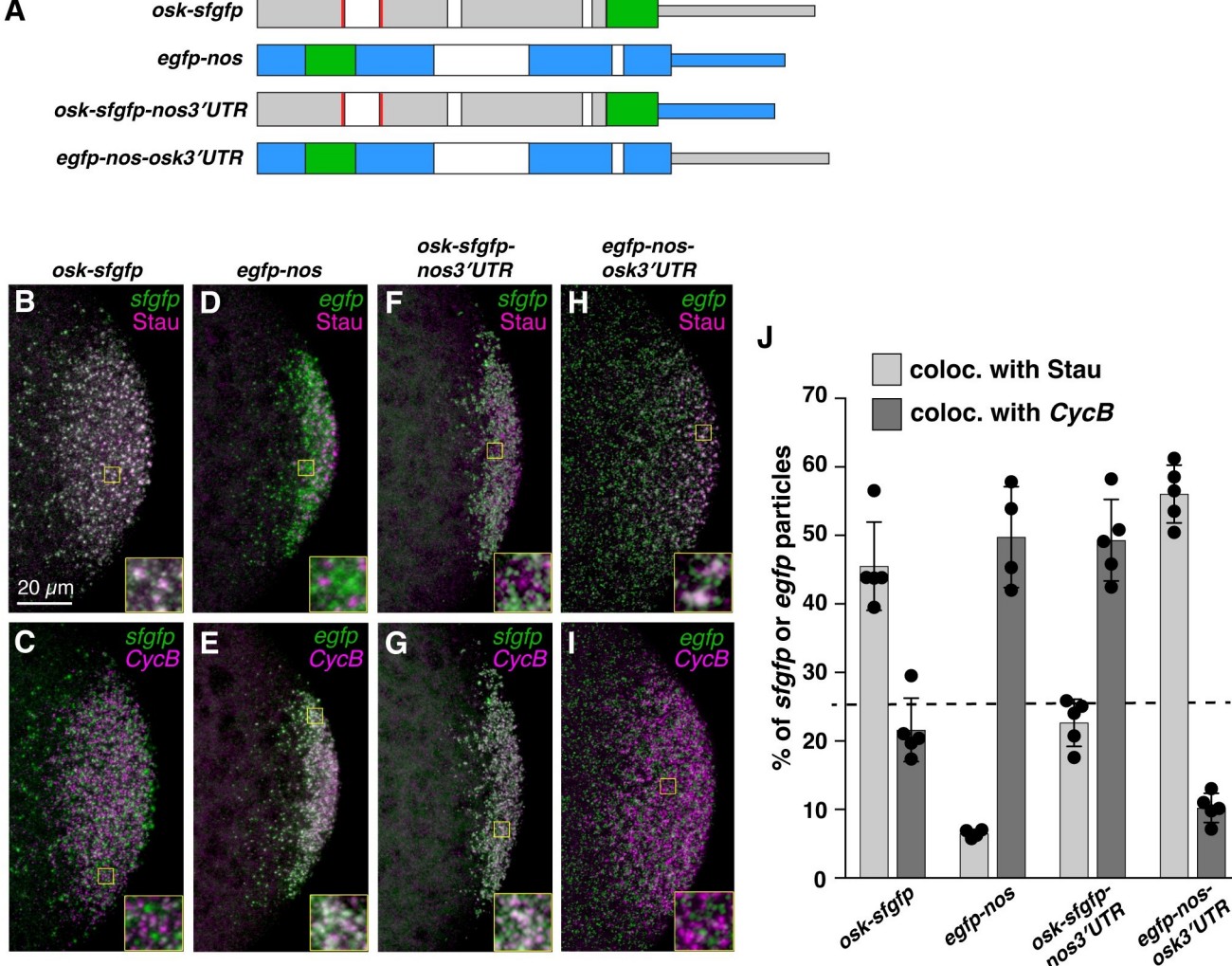

**Fig 6. The *osk* 3′ UTR is sufficient for founder granule targeting.** (A) Structure of transgenes. Gray boxes: *osk* exonic sequences; blue boxes: *nos* exonic sequences; white boxes: introns; red bars: sequences creating the SOLE; green boxes: *sfgfp* or *egfp*; thinner boxes indicate 3′ UTRs. (B-I) Confocal z-series projections of early embryos (≤ nuclear cycle 4). Anterior is toward the left. Transgenic mRNAs are detected by smFISH with probes for *sfgfp* or *egfp* (green). *CycB* smFISH (magenta) marks germ granules (C, E, G, I) and anti-Stau immunofluorescence (magenta) marks founder granules (B, D, F, H). Insets show enlargements of ROIs indicated by yellow boxes. Note that *CycB* is found in about 50% of all germ granules [18]. (J) Quantification of colocalization between transgenic mRNAs and either germ granules or founder granules; n = 4–5 embryos each. Individual data points and mean ± standard deviation are shown; p value for each pair <0.0001 as determined by 1-way ANOVA with Tukey's post-hoc test and by Student's t-test. Scale bars are indicated. Source data for the graphs in Fig 6J are provided in S1 Data.

the number of mRNAs per founder granule and the number of founder granules that contain the transgenic mRNA are reduced for *egfp-nos-osk3′UTR* as compared to *osk-sfgfp* mRNA (Fig 3K–3M). Total transgenic mRNA levels remain constant between these stages, however, suggesting that the loss of *egfp-nos-osk3′UTR* is unlikely a result of degradation (S4 Fig). Together, these data are most consistent with a failure to maintain *egfp-nos-osk3′UTR* in founder granules.

These results also suggest that the *osk* coding sequences or 5′ UTR contains an element important for maintenance of *osk* mRNA in founder granules for the duration of oogenesis. Since the *oskSOLE^{UA-mut}-sfgfp* mRNA is less enriched in the germ plasm of late oocytes than *osk-sfgfp* mRNA, we hypothesized that the SOLE could be such a maintenance element. In

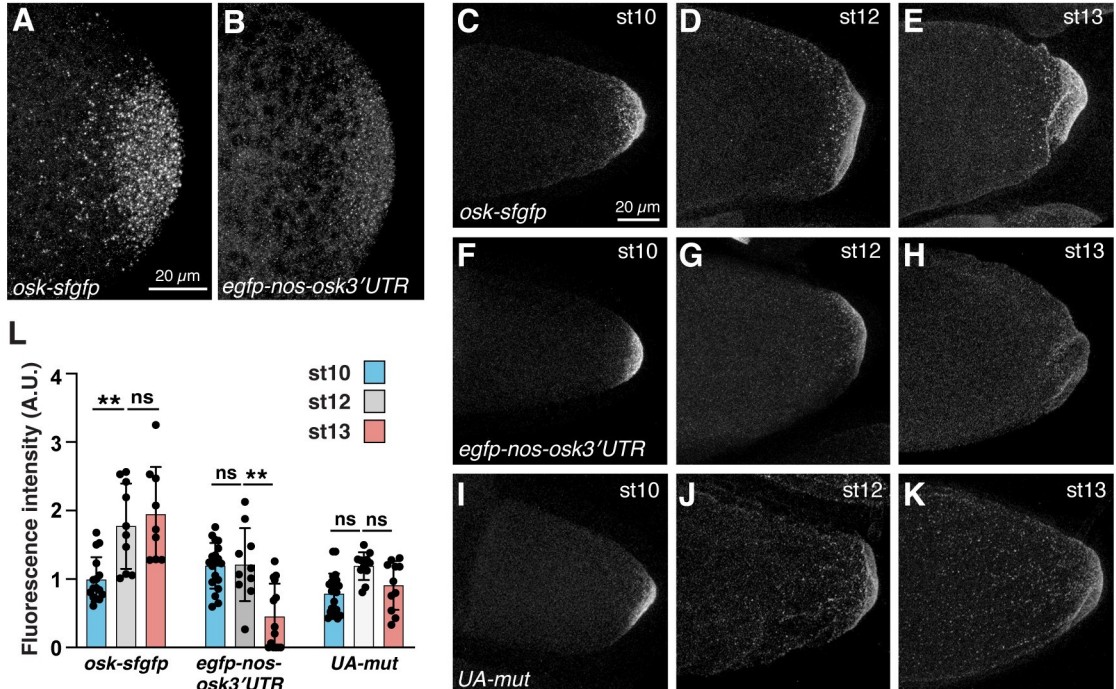

**Fig 7. Additional features are required to maintain *osk* after stage 12.** (A-B) Confocal z-series projections of early embryos showing the entirety of the germ plasm (nuclear cycle 3 to 5). (C-K) Confocal z-series projections of oocytes at stage 10 (C, F, I), stage 12 (D, G, J), and stage 13 (E, H, K) capturing the entirety of the germ plasm. Anterior is toward the left. Transgenic mRNAs were detected by smFISH with probes for *sfgfp* or *egfp*. (L) Quantification of fluorescence intensity of the germ plasm from stages 10 to 13; n = 9–22 oocytes each. Individual data points and mean ± standard deviation are shown; **p < 0.01 as determined by Kruskal-Wallis one-way analysis of variance, with Dunn's post-hoc test. Scale bars are indicated. Source data for the graphs in Fig 7L are provided in S1 Data.

contrast to *egfp-nos-osk3'UTR* mRNA, however, *oskSOLE*$^{UA-mut}$*-sfgfp* mRNA is not lost from the posterior after stage 12 (Fig 7I–7L). Thus, the SOLE and 3′ UTR most likely function in the association of *osk* RNPs to form founder granules whereas long-term persistence requires additional coding region or 5′ UTR sequences.

## Discussion

Our finding that founder granules appear to be agglomerates of *osk* RNPs provides new insight into the process by which *osk* accumulates at the posterior of the oocyte during late stages of oogenesis. *osk* is transported in RNPs containing 2–4 transcripts [18]. Recent work has shown that these RNPs are initially liquid-like condensates but they rapidly mature to a non-dynamic, solid state that prevents incorporation of additional mRNA molecules. Inducing a more liquid-like state results in formation of large, dynamic condensates at the posterior of late-stage oocytes that subsequently detach [19] indicating that the solid state is necessary for proper founder granule assembly and anchoring. Our observation that founder granules contain multiple physically distinct *osk* RNPs packed together is consistent with the solid-like properties of these RNPs and indicates that they do not form through the collapse of transport RNPs into larger condensates but rather through an aggregative process. This mechanism contrasts with *Drosophila* germ granules, whereby pre-formed protein condensates are populated by RNPs containing single transcripts, which then self-assemble within the granules to form homotypic clusters [40]. What limits the agglomeration of *osk* RNPs into founder granules to the posterior

of the oocyte remains unclear. RNP-RNP associations may be fostered by the high posterior concentration of *osk* RNPs achieved previously by kinesin-dependent transport. Since proteins can partition into *osk* RNPs after their transition to a solid-like state [19], proteins recruited to the germ plasm, perhaps by Osk protein itself, could mediate this behavior. Intriguingly, zebrafish germ plasm mRNAs form homotypic RNPs that aggregate into compact structures while retaining their distinct spherical appearance [47]. This similarity with founder granules suggests that agglomeration may be a more generalized mechanism for mRNA compartmentalization.

We interrogated the function of the earlier acting EJC/SOLE complex in late-phase *osk* localization and found that the SOLE, but not the adjacent EJC, is required. Whereas the function of the SOLE in the earlier localization phase relies only on the structure of the proximal stem [15], both the sequence of the proximal stem and its structure are important for late-phase localization. How the SOLE collaborates with the EJC to promote kinesin-dependent *osk* motility and what, if any regulatory factor interacts with it are not yet known. The sequence-dependence of the SOLE and lack of requirement for the EJC in late-phase localization suggests a different mode of action, possibly through the binding of a different protein to the proximal stem and recruitment of new RNP components or through RNA-RNA interactions. The change in ovarian physiology with the onset of nurse cell dumping could lead to an exchange of proteins associated with *osk*, to inhibit kinesin-dependent motility and promote posterior agglomeration.

The failure of *osk-K10_3'UTR* mRNA to localize at late stages of oogenesis despite the presence of the SOLE indicates that similarly to the earlier phase, late-phase localization depends on both the SOLE and the 3' UTR. This dependence on the 3' UTR for the late accumulation of *osk* is not for the purpose of hitchhiking, and by swapping the *osk* and *nos* 3' UTRs we showed that the *osk* 3' UTR specifies association of *osk* RNPs in founder granules independently of the SOLE. This function of the *osk* 3' UTR may be conferred by the same 3' UTR-binding proteins that control the formation and/or initial localization of *osk* transport RNPs and remain associated with *osk* at the posterior pole, such as Bru1, Stau, or Hrp48. For example, a prion-like domain in Bru1 required for formation of *osk* transport RNPs [19] could also mediate self-association of these RNPs when they come in contact at the posterior pole. Likewise, mammalian Stau has the propensity to form cytoplasmic aggregates [48]. The requirements for Bru1 and Stau in the earlier phase of *osk* localization [1, 2, 19] make it difficult to test this idea, however. Additionally, the *osk* 3' UTR may function to prevent co-condensation of *osk* RNPs with germ granules through the recruitment of proteins like Hrp48, which maintains the solid-like properties of *osk* RNPs [19].

Results from swapping the *osk* and *nos* 3' UTRs also suggest that either *osk* 5' UTR and/or coding sequences other than the SOLE contribute to maintaining *osk* RNPs in founder granules. Since multivalent interactions are typically required for inclusion of components in phase separated condensates [49], it is not surprising that binding of founder granule components to multiple sites within *osk* would be required for the integrity of these granules. Further dissection of the sequence requirements and identification of interacting factors will be necessary to define the mechanisms by which the various *osk* elements accomplish the different tasks.

The process by which *osk* mRNA achieves its posterior localization is remarkably complex and labor intensive, involving distinct machineries for transport into the oocyte, movement to the posterior pole during stages 8 to 10, and further accumulation during late stages of oogenesis. Given the dependence of embryonic abdominal patterning and germ cell formation on the amount of *osk* mRNA localized during oogenesis [30, 41, 42], the reliance on numerous distinct contributions to *osk* localization likely provides robustness to processes governing the targeting of *osk* RNPs to the right location and the accumulation of sufficient *osk* there.

Moreover, the distinct process of assembling founder granules ensures that *osk* mRNA remains separated from germ granules to promote its degradation in the embryonic germ plasm and minimize its inheritance by pole cells.

## Materials and methods

### Construction of transgenes and transgenic lines

The *osk-sfgfp* and *oskΔi1,2-sfgfp* transgenes and transgenic lines were previously described and contain *sfgfp* sequences inserted just before the *osk* stop codon in an 8 kb genomic *osk* rescue fragment in the pattB vector [43]. *oskΔi1,2-sfgfp* lacks the first and second *osk* introns [43]. The *egfp-nos* transgenes and transgenic lines were previously described and contain *egfp* sequences inserted just after the *nos* start codon in a 4.3 kb *nos* genomic rescue fragment [50]. To generate *osk-sfgfp-nos3′UTR*, the *osk* 3′ UTR and 3′ genomic DNA were removed from the plasmid pattB-osk-sfgfp [43] and replaced with a 1.3 kb fragment containing the *nos* 3′ UTR and 451 bp of 3′ genomic *nos* DNA. To generate *egfp-nos-osk3′UTR*, a fragment containing the *nos* promoter and 5′ UTR, *egfp*, and *nos* coding region (including introns) was removed from pCaS-peR-Pnos-gfp-nos [50], fused to the *osk* 3′ UTR and 3.2 kb of *osk* 3′ genomic sequences, and cloned into pattB. To generate the *osk-sfgfp* SOLE mutant transgenes, a 774 bp SphI-SacI fragment from pattB-osk-sfgfp [43] was replaced with a 774 bp SphI-SacI fragment synthesized by Genewiz with either the PS-Lz, UA$^{mut}$, or UA$^{mut-comp}$ mutation shown in Fig 3A. *UASp-osk-sfgfp-K10_3′UTR* was generated by inserting a fragment from pattB-osk-sfgfp [43] spanning a naturally occurring BamHI site just before the *osk* transcription start site to an engineered BamHI site immediately following the *osk* stop codon into the BamHI site of pattB-UASp. Transgenes were integrated into the attP40 site by phiC31-mediated recombination.

### Tissue collection and fixation

*Ovaries*: Females were fed on yeast paste at 25˚C for 3 days. Ovaries were dissected into PBS, lightly teased apart, and fixed and stepped into methanol as previously described [51]. Ovaries were stored in methanol at -20˚C for ≤1 month. Ovaries used for RNA extraction and RT-qPCR were transferred to 1.5 mL Eppendorf tubes after dissection, flash frozen in liquid nitrogen, and stored at -80˚C. *Embryos*: Embryos were collected on apple juice agar plates at room temperature, then dechorionated, fixed, and devitellinized as described [51]. Embryos were stored in methanol at -20˚C for ≤1 month. Embryos used for RNA extraction and RT-qPCR were dechorionated, flash frozen in liquid nitrogen, and stored at -80˚C.

### Single molecule fluorescence in situ hybridization (smFISH)

smFISH was performed according to Abbaszadeh and Gavis [51]. smFISH probe sets consisting of 20 nt oligonucleotides were designed with Stellaris Probe Designer and synthesized by Biosearch Technologies. Probes complementary to *egfp* (32 oligos) were conjugated to Atto 647N dye (Sigma-Aldrich) and purified by HPLC as previously described [52]. smFISH probes for *sfgfp* (31 oligos) were purchased already conjugated to Quasar 670 fluorophore from Biosearch Technologies. Probes were labeled with far-red fluorophores to avoid detecting fluorescence from Osk-sfGFP and Nos-EGFP proteins. For quantification of total localized fluorescence intensity, 1 μL of probes per 100 μL of hybridization buffer was used; for colocalization analysis and particle quantification, 3 μL of probes per 100 μL of hybridization buffer was used. Ovaries or embryos samples were mounted under #1.5 glass coverslips (VWR) in Vectashield Mountant (Vector Laboratories) for quantification of total localized fluorescence intensity or in Prolong Diamond Antifade Mountant (Thermo Fisher Scientific) for particle quantification.

## Immunofluorescence

Immunofluorescence was performed as previously described [43]. Embryos were incubated in rabbit ant-Staufen #36.2 (kindly provided by D. St Johnston) diluted 1:2000 in PBHT (PBS, 0.1% Tween-20, 0.25 mg/ml heparin [Sigma-Aldrich], 50 µg/ml tRNA [Sigma-Aldrich]) overnight at 4°C with rocking. Alexa-488 goat anti-rabbit secondary antibody (Molecular Probes) was diluted 1:1000 in PBHT and applied for 2 hr at room temperature with rocking. Embryos were mounted as described above. For double immunofluorescence/smFISH experiments, immunostaining was performed first, then embryos were refixed in 4% PFA for 30 min at room temperature with rocking and rinsed 4× with PBST (PBS, 0.1% Tween-20) before proceeding with smFISH as described above. Embryos were mounted in Prolong Diamond Antifade Mountant.

## Nile Red staining

Ovaries were fixed as described above but not treated with methanol. Ovaries were washed 2×5 min. with PBST and 2×10 min. with BBT (PBST, 0.1% globulin-free BSA), then incubated in 2 mg/ml Nile Red in BBT for 20 min. Ovaries were rinsed once with PBST, then mounted in Prolong Diamond Antifade Mountant.

## Microscopy and image quantification

Confocal imaging for experiments shown in Figs 2, 3, 6 and 7 was performed using a Leica SP5 laser scanning microscope with a 63× 1.4 NA oil immersion objective and GaAsP "HyD" detectors. For the smFISH experiment in Fig 5, confocal imaging was performed using a Nikon A1 microscope with a 40x 1.3NA oil immersion objective and GaAsP detectors. All imaging parameters were kept identical within each experiment. For experiments in Figs 1, 4, S1, and S2, confocal and 2D-STED imaging was performed on a Nikon A1R-STED with a 100x oil immersion objective. The *sfgfp* Quasar 670 probes were detected with the STAR RED setting. Nile Red was detected using the STAR ORANGE setting. Deconvolution of STED images was performed using the Nikon NIS-Elements built-in deconvolution tool.

For quantification of total fluorescence intensity, z-series with a 2 µm step size were used to capture the entire germ plasm-localized signal. Image processing and analysis were done in Fiji [53]. Z-projections were made with the "sum slices" function and the threshold adjusted so the entire localized signal was included. The total fluorescence intensity of the localized signal (integrated density function in Fiji) was then measured. For the time series analysis, an ROI was drawn to encompass the germ plasm and fluorescence intensity was measured for the ROI, then the ROI was moved to an anterior region of the oocyte and background fluorescence intensity was measured and subtracted from the germ plasm ROI measurement.

For quantification of localized mRNA particle number and size (number of mRNAs per particle), HyD detectors were used in photon counting mode and z-series covering half of the thickness of an oocyte or embryo were captured with a 340 nm step size. Particles were identified and quantified as previously described [40] using a threshold of 0.5 for *sfgfp* probes and 0.75 for *egfp* probes. Only particles containing ≥4 mRNAs were included in the quantification [18].

Data are displayed as mean ± standard deviation. For 2-sample comparisons, the Student's t-test was used; for multiple comparisons, a one-way ANOVA was used with either Tukey or Dunnett's post-hoc test as indicated in the figure legends. Statistical analysis was performed using GraphPad Prism software.

## RT-qPCR

RNA was extracted from dechorionated embryos using the RNeasy kit (Qiagen). 0.75 µg total mRNA was used to generate cDNA using the Quantitect RT kit (Qiagen). 2 µl cDNA was combined with 25 µl 2× TaqMan Gene Expression Master Mix (Thermo Fisher Scientific), 2.5 µl of 20× TaqMan Gene Expression Assay (Thermo Fisher Scientific, *sfgfp* custom–APT2CRZ, 4331348, *egfp* Mr 04097229_mr Enhance, or *rpl7* Dm 01817653, 4351372), and 20.5 µl of nuclease free $H_2O$. qPCR was performed on an Applied Biosystems 7900HT standard 96-well qPCR instrument. Three biological replicates were performed with three technical replicates each, all using a CT threshold of 0.6613619. Technical replicates were averaged and the three biological replicates were normalized to the *rpl7* control using the $\Delta C_t$ method and presented as mean ± standard deviation. Statistical significance was determined by Student's t-test or by one-way ANOVA and Tukey's multiple comparisons tests, as indicated in the figure legends, using GraphPad Prism software.

## Supporting information

**S1 Data. Original data accompanying Figs 2, 3, 6, 7, S2 and S4.** Individual values used to generate graphs for each figure are listed under the corresponding tab.
(XLSX)

**S1 Fig. 2D-STED analysis of *osk* mRNA particles during stages 8 to 9 of oogenesis.** Images were taken of the nurse cells, the anterior region of the oocyte, and the posterior cortex of the oocyte in wild-type egg chambers at stage 8 (A) and stage 9 (B). *osk* mRNA was detected by smFISH. Panels in the top row are single confocal sections. The yellow boxes indicate ROIs imaged using STED microscopy as shown in the panels in the bottom rows. Panels in the middle rows show confocal images of each ROI prior to STED for comparison. Images are rendered using the Red Hot lookup table in Fiji and scale bars are indicated.
(TIF)

**S2 Fig. 2D-STED control experiments.** (A) Images of the posterior cortex of wild-type stage 13 oocytes. *osk* mRNA was detected by smFISH. Panels in the top row are single confocal sections. The yellow boxes indicate ROIs imaged using STED microscopy as shown in the panels in the bottom row. Panels in the middle row show confocal images of each ROI prior to STED for comparison. (B) Image of the posterior region of a stage 13 oocyte stained with Nile Red to detect lipid droplets. Confocal and STED images of the ROI indicated by the yellow box are shown in the middle and lower panels, respectively. Images are rendered using the Red Hot lookup table in Fiji and scale bars are indicated.
(TIF)

**S3 Fig. Quantitation of transgenic mRNA levels.** RT-qPCR quantification of transgenic mRNA extracted from early embryos, normalized to *rpl7* mRNA. Individual data points and mean ± standard deviation are shown. Values are not significantly different from the *osk-sfgfp* control, as determined by one-way ANOVA and Dunnett's multiple comparisons test. Source data for the graphs in S3 Fig are provided in S1 Data.
(TIF)

**S4 Fig. Stability of transgenic mRNAs during late stages of oogenesis.** Fluorescence intensity measurements of unlocalized *osk-sfgfp*, *egfp-nos-osk3′UTR* and *osk-sfgfpSOLE^UA-mut* mRNAs in stage 12 and stage 13 oocytes; n = 11–14 oocytes each. Individual data points and mean ± standard deviation are shown. The values are not statistically significant as determined by one-way ANOVA and Tukey's post-hoc test. Source data for the graphs in S4 Fig are

provided in S1 Data.
(TIF)

## Acknowledgments

We are grateful to D. St Johnston for anti-Stau antibody and A. Ephrussi and P. Macdonald for plasmid DNA. We thank G. Laevsky and S. Wang in the Princeton Confocal Imaging Facility, a Nikon Center of Excellence in the Department of Molecular Biology, for assistance with microscopy, S. Chatterjee for technical assistance, M. Niepielko for assistance with particle quantification, and A. Hakes and D. Bolton for comments on the manuscript.

## Author Contributions

**Conceptualization:** Catherine E. Eichler, Elizabeth R. Gavis.

**Formal analysis:** Catherine E. Eichler, Hui Li, Elizabeth R. Gavis.

**Investigation:** Catherine E. Eichler, Hui Li, Michelle E. Grunberg, Elizabeth R. Gavis.

**Project administration:** Elizabeth R. Gavis.

**Supervision:** Elizabeth R. Gavis.

**Writing – original draft:** Catherine E. Eichler.

**Writing – review & editing:** Hui Li, Michelle E. Grunberg, Elizabeth R. Gavis.

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
