## [Decision Letter · Decision Letter 0]

8 May 2023

Dear Liz,

Thank you very much for submitting your Research Article entitled 'Localization of oskar RNA by agglomeration in ribonucleoprotein granules' to PLOS Genetics.

The manuscript was fully evaluated at the editorial level and by independent peer reviewers. The reviewers appreciated the attention to an important topic but identified some concerns that we ask you address in a revised manuscript. In particular, Reviewer 1 suggests some interesting technical control experiments for STED; and reviewer 2 makes some recommandations to make the manuscript more accessible to non-specialist readers. 

We therefore ask you to modify the manuscript according to the review recommendations. Your revisions should address the specific points made by each reviewer.

Yours sincerely,

Jean-René Huynh

Academic Editor

PLOS Genetics

Gregory P. Copenhaver

Editor-in-Chief

PLOS Genetics

Reviewer's Responses to Questions

**Comments to the Authors:**

Reviewer #1: The manuscript by Eichler et al. describes efforts to understand the processes that govern the assembly of oskar RNP granules at the posterior of the Drosophila oocyte during late oogenesis. Localization of oskar at this site plays a critical role in germline development and axial patterning. Whilst oskar trafficking in mid-oogenesis has long served as a paradigm for mRNA localization and translational control, relatively few studies have addressed the mechanisms underpinning the posterior localization of this molecule in late-stage oocytes. Therefore, the manuscript addresses an important problem. Building on their previous studies that assessed the composition of granules in the late-stage oocyte, the authors now address the process by which oskar RNPs increase in size to form ‘founder granules’ (structures which they have previously shown mediate compartmentalized degradation of oskar). The current study presents evidence from superresolution microscopy that founder granules form by agglomeration of smaller RNPs and that the set of features that drive their formation are distinct from those used to drive posterior localization in mid-oogenesis. These latter experiments involve dissection of cis-acting elements through a series of transgenic reporter constructs. Overall, the experiments are well controlled, the images are convincing and the effects are quantified robustly. The manuscript is also very well written. The study is a nice illustration of how a single mRNA can use different localization mechanisms in different contexts. I am supportive of publication but have a few minor suggestions for strengthening the manuscript.

1. The images in Figure 1 certainly seem to support an agglomeration process acting by stage 13. However, the conclusion would be more convincing if the authors could produce evidence that the punctate distribution of signals in the STED images is not an artefact of this imaging modality in the complex environment of the oocyte. For example, can the authors provide STED images of other structures (e.g. membrane compartments/nucleus) in late-stage oocytes that have a more uniform signal? And can the authors rule out that the punctate pattern stems from stochastic hybridization of probes within the founder granule (e.g., would the same signal be seen with a higher concentration of probe?).

2. The conclusion (made on multiple occasions) that agglomeration of oskar RNPs is independent of the EJC is, in this reviewer’s opinion, too strong based on the current data. They show that this process does not depend on the first two introns of oskar but it is formally possible that the EJC is required through a non-canonical mechanism. I recommend toning down this conclusion to state something along the lines of the EJC not appearing to be involved, at least not through the same process that occurs in mid-oogenesis.

3. In Figure 5C, does the K10 3’UTR enrich the RNA at the anterior of the oocyte? If this is the case, it may be that agglomeration is not occurring in this genotype because on an indirect effect of having too low a concentration of oskar RNA in the posterior region. I recognize that the conclusions are subsequently extended by the nos construct but nonetheless I recommend considering the addition of a caveat when introducing the K10 data.

4. Figure 3 legend, panel J: should these read ‘at stages 10 and 13’ instead of ‘from 10 to 13’?

5. Figure 7 legend: There is a typo in the line beginning ‘Transgenic RNAs were detected…’ (‘using with’).

6. There is a figure legend for Figure S4 but it does not seem this figure is called out in the manuscript.

7. Figure 6. The authors may wish to consider color blind readers and change the colour scheme in panels B to I to green and magenta.

Reviewer #2: The process of oskar mRNA delivery to the posterior of the oocyte has been in the focus since decades. However, the accumulation of large osk mRNA granules is still not well understood. It is in this last step that the authors apply super-regulation microscopy and shed new light on this recruitment of granules at the posterior that is responsible for morphological and functional asymmetry. Here the Gavis lab has contributed significantly in the recent past, showing that osk RNA multimerizes together with Bruno to form translationally repressed complexes. A first key finding is that, in contrast to the first part of the process, late phase osk mRNA localization is independent of the EJC. In contrast, the previously identified SOLE element in the RNA is still required for the accumulation process. Later experiments presented in Fig. 6, however, seem to suggest that “the SOLE is not sufficient to target RNAs to founder granules, rather the osk 3’-UTR”. Next, the authors report that it is the osk coding and 5’-UTR, but not the SOLE that seem to be required to maintain osk RNA accumulation.

Overall, the quality of the presented data is impressive and convincing. However, at least for a person not working on osk RNA, it is difficult to follow the experimental logic between the different blocks (see above) of the results. In my opinion, a graphical representation of osk RNA with its different elements and their positions and possible interaction(s) might really help. Secondly, it would be helpful to integrate the data into an RNA-centric osk model of how different signals might drive different phases of localization. As the authors mention the different contributions of dynein and kinesin, do they have data whether any of the motors contribute to the late accumulation? At least, it would be very informative to integrate motors into the working model.

Specific points for the authors:

• Figure 1: Though the authors are limited by 2D STED, it would be great to get an initial quantification of this data, e.g. changes in size, number of RNA molecules in agglomerates, or a quantification of clustering. As this is one of the core aspects of the paper, a quantification should be attempted in all figures where agglomerates are investigated (Fig. 4).

• For purpose of clarity in line 200, the authors might indicate that both structure and sequence are required for late-phase accumulation. The current wording is contrived and unclear.

• In line 209, the text refers to a quantification in Fig. 4K,L, which is not part of the figure. It would seem the authors meant Fig.3. However, as mentioned above, a quantification of STED images would be encouraged, to clarify this point. The authors conclude, that SOLE mutants are unable to accumulate in late-stage founder granules. Would such a failure not increase the number of individual granules present, quantified in Fig. 3K? As mRNA levels remain constant, is the reporter mRNA localized to a different area?

• The reasoning in line 252 is unclear, could the authors specify. How was Stau2 488 signal separated from GFP? Was GFP/488 signal observed in cycB granules? If osk-GFP or Stau2-488 was unspecific in germ granules, how can they be specific in founder granules? Please provide a clearer explanation. It is a nice idea to combine these aspects in the same embryos. However, perhaps it would have made more sense to perform this analysis in different embryos to ensure spectral separation.

• Please provide magnified insets for the images in all figures, as already done for the STED figures, to clearly show the granules.

• Overall, do not use too many distinct terms for granules, which might be somewhat confusing for the general audience, e.g. RNPs, granules, bodies, etc.

In conclusion, I am convinced that this topic is important, original and timely and I give the Gavis lab credit for important new insights into RNA localization. With these additional pieces of information, I anticipate a better and more thorough understanding of how SOLE and different elements within osk RNA might contribute to sequential localization of osk RNA.

**Have all data underlying the figures and results presented in the manuscript been provided?**

Reviewer #1: **No: **I should not see a spreadsheet for the plotted data in this submission. It should be easy to add if the m/s is accepted.

Reviewer #2: Yes

PLOS authors have the option to publish the peer review history of their article (what does this mean?). If published, this will include your full peer review and any attached files.

Reviewer #1: No

Reviewer #2: No

---

## [Editor Report · Decision Letter 1]

19 Jul 2023

Dear Dr Gavis,

We are pleased to inform you that your manuscript entitled "Localization of oskar RNA by agglomeration in ribonucleoprotein granules" has been editorially accepted for publication in PLOS Genetics. Congratulations!

Yours sincerely,

Jean-René Huynh

Academic Editor

PLOS Genetics

Gregory Copenhaver

Editor-in-Chief

PLOS Genetics

Comments from the reviewers (if applicable):

**Data Deposition**

http://datadryad.org/submit?journalID=pgenetics&manu=PGENETICS-D-23-00330R1

**Press Queries**

---

## [Editor Report · Acceptance letter]

21 Aug 2023

PGENETICS-D-23-00330R1 

Localization of oskar RNA by agglomeration in ribonucleoprotein granules 

Dear Dr Gavis, 

We are pleased to inform you that your manuscript entitled "Localization of oskar RNA by agglomeration in ribonucleoprotein granules" has been formally accepted for publication in PLOS Genetics! Your manuscript is now with our production department and you will be notified of the publication date in due course.

With kind regards,

Olena Szabo

PLOS Genetics

On behalf of:
